# Binge drinking and associated factors among school students: a cross-sectional study in Zhejiang Province, China

Hao Wang,[1] Ruying Hu,[1] Jieming Zhong,[1] Huaidong Du,[2,3] Bragg Fiona,[2] Meng Wang,[1] Min Yu[1]

[1]Department of NCDs Control and Prevention, Zhejiang Provincial Center for Diseases Control and Prevention, Hangzhou, China
[2]Clinical Trial Service Unit & Epidemiological Studies Unit (CTSU), Nuffield Department of Population Health, University of Oxford, Oxford, UK
[3]Medical Research Council Population Health Research Unit, Nuffield Department of Population Health, University of Oxford, Oxford, UK

**Correspondence to**
Professor Min Yu;
myu@cdc.zj.cn

## ABSTRACT

**Objective** To investigate the prevalence and correlating factors of binge drinking among middle and high school students in Zhejiang Province, China.

**Methods** We performed a cross-sectional study using data from a school-based survey. A total of 23 543 (response rate=97.5%) eligible adolescents from 442 different schools (including middle schools, academic high schools and vocational high schools) were asked to fill in an anonymous self-administered behaviour questionnaire between April and May 2017. Multivariable logistic regression models were used to examine the associations of sociodemographic and behavioural factors with binge drinking.

**Results** The mean (SD) age of participants was 15.6 (1.7) years and 51.3% were boys. The proportions of students from middle schools, academic high schools and vocational high schools were 51.9%, 27.5% and 20.6%, respectively. In total, 22.8% (95% CI 21.6 to 23.9) of students reported drinking alcohol in the past 30 days and 9.2% (95% CI 8.5 to 10.0) of students reported binge drinking (defined as drinking four or more alcoholic drinks in 1–2 hours period among girls and five or more alcoholic drinks among boys) during the past month. The prevalence of binge drinking was highest among vocational high school students (17.9% vs 6.3% and 7.7% among middle school and academic high school students, respectively). Older age, studying at high school, poor academic performance, higher levels of physical activity, excessive screen-time, loneliness, insomnia, previous suicide attempt, cigarette smoking, fighting, being bullied and sexual experience were found to be positively associated with adolescent binge drinking.

**Conclusions** Binge drinking is common among middle and high school students in Zhejiang, China. Efforts to prevent binge drinking may need to address a cluster of sociodemographic and behavioural factors. Our findings provide information to enable healthcare providers to identify students at high-risk of binge drinking and to inform planning of intervention measures for at-risk students.

## INTRODUCTION

Alcohol consumption has been identified as an important risk factor for chronic disease and injury.[1] Disability-adjusted life-years attributable to alcohol consumption have

### Strengths and limitations of this study

► This is a school-based study with a relatively large sample size, a high response rate and a study population which is representative of Provincial China.
► The study questionnaire includes a wide range of factors, which might help us better understand how to prevent adolescent binge drinking.
► The cross-sectional study design prevents establishment of causal relationships between sociodemographic or behavioural factors and binge drinking.
► Only students attending schools participated in the survey. Students who have been expelled or suspended from school, or who stopped attending, may be more likely to binge drink, and the overall prevalence of binge drinking in our current study might therefore represent an underestimate of the true prevalence.

increased by >25% between 1990 and 2016, accounting for an estimated 99.2 million worldwide. In 2016, alcohol consumption was responsible for >2.8 million deaths worldwide, including approximately 0.7 million in China.[2] Many studies have clearly demonstrated that alcohol misuse among adolescents is associated with motor vehicle collisions,[3] violence and crime,[4 5] mental health disorders and higher risk of suicide.[6–8] Moreover, alcohol use during early adolescence is a risk factor for later alcohol dependence.[9–11]

Binge drinking is defined as episodic excessive drinking. Prevalence of binge drinking among adolescents varies between countries, and is estimated to be 8% in Iceland and 56% in Denmark.[12] Among 25 European countries, prevalence of binge drinking among adolescents rose from 36% in 1995 to 42% in 2007, and then declined to 35% in 2015.[12] In the USA, the prevalence declined from 31.3% in 1991 to 17.7% in 2015.[13] In Asia, the prevalence of heavy episodic drinking (ie, binge drinking) among grade 7–12 students was 10.3% in Korea,[14] while 21.2% of students, with mean age 14.7 years, reported

drinking alcohol in the past 30 days and 7.1% reported binge drinking in Hong Kong.[15] In mainland China, a study of 7344 middle school students conducted in four cities (Beijing, Hangzhou, Wuhan, Urumchi) found that the prevalence of current drinking was 14.4%, and 9.9% of students had experienced drunkenness, with significant regional differences.[16] In another study, conducted in 2004 and including 54 040 students in grades 7–12 from 18 Provincial capitals in China, 25.2% of students reported consuming at least one alcoholic drink and 10.3% reported at least one episode of binge drinking during the 30 days prior to being surveyed.[17] A more recent study, surveying 13 811 high school students from Beijing, Shanghai and Guangzhou in 2013, found the prevalence of binge drinking was 4.8%,[18] although not all school grades were included, likely resulting in underestimation of the true prevalence.

Parental alcohol-related practices have been shown to be important risk factors for adolescent drinking.[19] In traditional Chinese culture, alcohol drinking is both a normal part of the daily diet, especially in rural areas, and an important part of rituals, business occasions, festivals and special events. Moderate drinking on important occasions is encouraged for adults,[20] and children may be given small amounts of alcoholic beverages by parents or grandparents, especially grandfathers.[18] Some previous surveys have found that nearly half of senior high schools students in China have drunk at home with their parents or outside the home with their peers.[21 22]

Previous studies have found some inconsistencies in sociodemographic patterns of adolescent binge drinking. For instance, Miller et al[23] reported that binge drinking prevalence was similar among high school boys and girls in the USA, while boys in Hong Kong had higher prevalence of binge drinking than girls.[15] Sokol-Katz et al[24] reported that family structure was not significantly related to behaviours such as alcohol, cigarette and drug use among middle school students in Florida, after controlling for gender and race/ethnicity, while growing up in a non-intact family was associated with substance use among Chinese adolescents in Hong Kong.[25]

Zhejiang Province, in the east of China, has a population of 56 million. It has experienced rapid economic development in the past 30 years, which has been associated with increased exposure of adolescents in Zhejiang to western culture. As a result of this, patterns of alcohol use in this area might differ from those in other regions of China. The current study was designed with the aim of examining the prevalence of binge drinking and its correlating factors among students in Zhejiang Province.

# METHODS
## Survey design and participants
The survey used a three-stage sampling design. In stage 1, 30 counties, including 12 urban areas and 18 rural areas, were sampled randomly from all 90 counties (31 urban and 59 rural) of Zhejiang Province according to

socioeconomic status. In stage 2, 10 classes of middle school, 5 classes of academic high school and 5 classes of vocational high school were selected randomly within each chosen counties, respectively. In stage 3, all students in the chosen classes were invited to participate in the study. In China, after 6 years of primary school education, children usually attend middle school (ie, junior high school) for 3 years (grades 7–9). After graduation from middle schools, they enter high school (ie, senior high school, including academic high schools and vocational high schools) for a further 3 years (grades 10–12). The survey questionnaire was modelled from existing surveys including the Youth Risk Behaviour Survey, conducted by the Centers for Disease Control and Prevention (CDC),[26] and the international Global School-based Student Health Survey, supported by WHO.[27] Survey questions addressed demographic characteristics (including age, gender, parental educational level, parental marital status, number of siblings), tobacco and alcohol use, physical activity, violence, injuries, suicidal behaviours and sexual behaviours. The survey was conducted between April and May 2017. Participants completed the anonymous, self-administrated questionnaire in the classroom.

## Sample size calculation
The sample size was calculated by using the formula: $n = deff \times \mu^2 \times P \times (1-P)/d^2$. Means and 95% CI (2-sided for $\mu = 1.96$) were determined; the prevalence of binge drinking (10%) obtained in the China was used as a measure of probability (P);[17] the design effect (deff) value was set at 3 and the relative error was: $d = r \times 1\%$, $r = 15\%$. Based on these parameters, the sample size for each stratum was estimated to be 4610 subjects. Because there were four strata (areas: urban and rural; sex: boy and girl), and assuming a potential non-response rate of 20%, the final sample size was calculated as 23 050.

## Patient and public involvement
Study participants were healthy students and no patients were involved in the study. Students and their parents were not involved in the design and conduct of study. Due to an anonymous survey, our findings will be disseminated to Department of Health and Department of Education, not directly to participating students.

## Measures
### Outcome variables
For the purposes of this study, alcoholic drinks include beer, wine, wine coolers, rice wine and liquor such as Chinese liquor, rum, gin, vodka or whiskey. Current drinking was assessed through the question: "During the past 30 days, on how many days did you have at least one drink of alcohol?" Answer options include: '0 days', '1–2 days', '3–5 days', '6–9 days', '10–19 days', '20–29 days' and '30 days'. Participants were considered as current drinkers if they reported drinking alcohol on at least 1–2 days during the past 30 days. Binge drinking was assessed by the question: "During the past 30 days, on how many

days did you have 4 or more drinks of alcohol (if you are girl) or 5 or more drinks of alcohol (if you are boy) during a period of 1 to 2 hours?"[28] Answer options include: '0 days', '1 day', '2 days', '3–5 days', '6–9 days', '10–19 days', '≥20 days'. Participants were defined as binge drinkers if they answered at least 1 day.

## Main covariates

Physical activity was assessed by the question: "During the past 7 days, on how many days were you physically active for a total of at least 60 min per day?" Answer options included: 'none', '1 day', '2 days', '3 days', '4 days', '5 days', '6 days' and '7 days', and these were further categorised into four groups: 'none', '1–2 days/week', '3–5 days/week' and '6–7 days/week'. Current smoking was assessed by the question: "During the past 30 days, on how many days did you smoke cigarettes?" Answer options included: 'none', '1–2 days', '3–5 days', '6–9 days', '10–19 days', '20–29 days' and 'all 30 days'. Current smoking was defined as smoking cigarettes on at least 1–2 days in the past 30 days. Screen-time was estimated through the question: "On an average school day, how many hours do you play video or computer games or use a computer for something that is not school work?" Answer options included: 'none', '<1 hour/day', '1 hour/day', '2 hours/day', '3 hours/day', '4 hours/day', '≥5 hours/day', and these answers were further categorised into four groups: 'none', '<1 hour/day', '1–4 hours/day', '≥4 hours/day'. Suicidal attempt was assessed using the question: "During the past 12 months, how many times did you actually attempt suicide?" Response options included: 'none', '1 time', '2–3 times', '4–5 times', '6 or more times', and suicide attempt was defined as at least once in the past 12 months. Fighting was assessed by the question: "During the past 12 months, how many times were you in a physical fight?" Answer options included: 'none', '1 time', '2–3 times', '4–5 times', '6–7 times', '8–9 times', '10–11 times', '12 or more times', and fighting was defined as at least once in the past 12 months. Being bullied was assessed by the question: "During the past 12 months, how many times has someone threatened or injured you with a weapon such as a gun, knife or club on school property?" Answer options included: 'none', '1 time', '2–3 times', '4–5 times', '6–7 times', '8–9 times', '10–11 times', '12 or more times'. Being bullied was defined as being threatened or injured by someone at least once in the past 12 months. More detailed information on covariates is provided in table 1.

## Quality control

The survey was conducted by trained surveyors from the local CDC. In order to improve response rate, every recruited student was given a pencil box as a gift, and the survey was anonymous.

## Ethics statement

Written informed consent was obtained from all participants and their guardians before the survey.

**Table 1** Questions comprising variables included in the survey

| Variables | Questions and options |
| --- | --- |
| Parental education level | What is the highest level of education your father/mother has obtained? (Answer options: primary school or below, middle school, high school, college or university, master graduates or above, unknown) |
| Parental marital status | What is your parents' current marital status? (Answer options: married, divorced, widowed, separated) |
| Siblings | Are you the only son/daughter of your parent? (Answer options: yes, no) |
| Physical activity | During the past 7 days, on how many days were you physically active for a total of at least 60 min per day? (Answer options: none, 1 day, 2 days, 3 days, 4 days, 5 days, 6 days, 7 days) |
| Academic performance | How would you describe your grades in class? (Answer options: excellent, middle, poor) |
| Screen-time | On an average school day, how many hours do you play video or computer games or use a computer for something that is not school work? (Answer options: none, <1 hour/day, 1 hour/day, 2 hours/day, 3 hours/day, 4 hours/day, ≥5 hours/day) |
| Loneliness | During the past 12 months, did you ever feel lonely? (Answer options: never, occasional, sometimes, often, always) |
| Insomnia | During the past 12 months, have you ever felt worried about something such that you cannot fall asleep? (Answer options: never, occasional, sometimes, often, always) |
| Suicidal attempt | During the past 12 months, how many times did you actually attempt suicide? (Answer options: none, 1 time, 2–3 times, 4–5 times, 6 or more times) |
| Current smoking | During the past 30 days, on how many days did you smoke cigarettes? (Answer options: none, 1–2 days, 3–5 days, 6–9 days, 10–19 days, 20–29 days and all 30 days) |
| Fighting | During the past 12 months, how many times were you in a physical fight? (Answer options: none, 1 time, 2–3 times, 4–5 times, 6–7 times, 8–9 times, 10–11 times, 12 or more times) |
| Being bullied | During the past 12 months, how many times has someone threatened or injured you with a weapon such as a gun, knife or club on school property? (Answer options: none, 1 time, 2–3 times, 4–5 times, 6–7 times, 8–9 times, 10–11 times, 12 or more times) |
| Sexual experience | Have you ever had sexual intercourse? (Answer options: yes, no) |

## Statistical analysis

All analyses were performed using SAS software V.9.3. A weighting factor was applied to each student record to

adjust for non-response and for the varying probabilities of selection. The weight used for estimation in this survey is given by: W=W1×W2×f1×f2. W1=the inverse of the probability of selecting the county; W2=the inverse of the probability of selecting the classroom within the county; f1=a student-level non-response adjustment factor calculated by class; f2=a poststratification adjustment factor calculated by grade.[29] Continuous variables were given as the mean±SD. The prevalence of current drinking and binge drinking was given as per cent and 95% CI. Weighted prevalence was calculated using the PROC SURVEYFREQ procedure and its difference between groups was compared using Rao-Scott $X^2$ test. To assess the associations between each correlating factor and binge drinking, univariate and multivariable logistic regression analyses were performed using the PROC SURVEYLOGISTC procedure, to take into account the complex survey sampling methods. We first determined which factors were associated with binge drinking in univariate analyses (P<0.05), and variables significant in the univariate analyses were entered in a multivariable logistic regression model. All statistical tests were two tailed, and P values <0.05 were considered to be statistically significant.

## RESULTS
### Descriptive statistics
A total of 24 157 students were invited to participate. Due to missing or incomplete questionnaires and refusal to participate, 23 543 eligible participants (response rate 97.5%) were included in the current analyses. Twelve thousand sixty-eight (51.3%) participants were boys and, overall, mean age was 15.6 years. Twelve thousand two hundred seven (51.9%) participants were middle school students, 6477 (27.5%) were academic high school students and 4859 (20.6%) came from vocational high schools.

The percentage of students that came from non-intact families was 9.9 (table 2); 11.4% of students' paternal educational level was college or above; 17.7% of students' maternal educational level was high school; 54.8% of students had siblings; 22.5% of students reported having excellent academic performance; 20.7% of students reported being physically active 6–7 days per week; 21.5% of students reported >4 hours screen-time per day; 64.1% of students reported never or occasionally feeling lonely during the past 12 months; 4.7% of students reported being often or always worried about something such that they could not sleep during the past 12 months; 5.5% of students reported smoking cigarettes during the past 30 days; 15.6% of students reported engaging in a physical fight and 13.2% reported being bullied during the past 12 months and 3.9% of students reported ever having sexual intercourse.

### The prevalence of current drinking and binge drinking
As shown in table 3, the prevalence of current drinking was 22.8% (95% CI 21.6 to 23.9), higher among boys than girls

**Table 2** Characteristics of adolescents from Zhejiang (n=23 543)

| Characteristics | Total | Boys | Girls |
|---|---|---|---|
| Age range (years) | | | |
| ≤13 | 5159 (20.9) | 2689 (21.1) | 2470 (20.8) |
| 14 | 4300 (17.8) | 2192 (17.7) | 2108 (17.8) |
| 15 | 3730 (16.5) | 1905 (16.5) | 1825 (16.6) |
| ≥16 | 10 354 (44.8) | 5282 (44.7) | 5072 (44.8) |
| Area | | | |
| Urban | 9022 (31.8) | 4544 (31.8) | 4478 (31.9) |
| Rural | 14 521 (68.2) | 7524 (68.2) | 6997 (68.1) |
| Types of school | | | |
| Middle school | 12 207 (51.8) | 6364 (52.4) | 5843 (51.0) |
| Academic high school | 6477 (26.1) | 3223 (25.1) | 3254 (27.3) |
| Vocational high school | 4859 (22.1) | 2481 (22.5) | 2378 (21.7) |
| Parental marital status | | | |
| Married | 21 151 (90.1) | 10 924 (90.9) | 10 227 (89.2) |
| Others | 2392 (9.9) | 1144 (9.1) | 1248 (10.8) |
| Paternal education level | | | |
| Middle or below | 13 568 (60.0) | 6908 (59.5) | 6660 (60.5) |
| High school | 5100 (20.8) | 2628 (20.8) | 2472 (20.7) |
| College or above | 3129 (11.4) | 1575 (11.2) | 1554 (11.7) |
| Unknown | 1746 (7.8) | 957 (8.5) | 789 (7.1) |
| Maternal education level | | | |
| Middle or below | 14 530 (63.9) | 7292 (62.5) | 7238 (65.5) |
| High school | 4363 (17.7) | 2271 (18.0) | 2092 (17.3) |
| College or above | 2736 (10.0) | 1392 (9.8) | 1344 (10.2) |
| Unknown | 1914 (8.4) | 1113 (9.7) | 801 (7.0) |
| Having siblings | | | |
| Yes | 12 137 (54.8) | 5263 (46.6) | 6874 (63.8) |
| No | 11 406 (45.2) | 6805 (53.4) | 4601 (36.2) |
| Academic performance | | | |
| Excellent | 5448 (22.5) | 2731 (21.7) | 2717 (23.3) |
| Middle | 11 765 (50.5) | 5727 (48.1) | 6038 (53.1) |
| Bad | 6330 (27.0) | 3610 (30.2) | 2720 (23.6) |
| Physical activity (days/week) | | | |
| 0 | 4883 (21.5) | 2079 (18.1) | 2804 (25.2) |
| 1–2 | 5690 (24.8) | 2703 (22.9) | 2987 (26.8) |
| 3–5 | 8050 (33.0) | 4237 (34.5) | 3813 (31.5) |
| 6–7 | 4920 (20.7) | 3049 (24.5) | 1871 (16.5) |
| Screen-time (hour(s)/day) | | | |
| 0 | 7255 (31.2) | 3604 (29.7) | 3651 (32.8) |
| <1 | 4009 (15.9) | 1968 (15.1) | 2041 (16.7) |
| 1–4 | 7621 (31.4) | 3853 (31.1) | 3768 (31.6) |
| ≥4 | 4658 (21.5) | 2643 (24.1) | 2015 (18.9) |
| Loneliness | | | |
| Never/occasional | 15 122 (64.1) | 8082 (66.8) | 7040 (61.2) |
| Sometimes | 5783 (24.7) | 2698 (22.6) | 3085 (26.8) |
| Often/always | 2638 (11.2) | 1288 (10.6) | 1350 (12.0) |
| Insomnia | | | |
| Never/occasional | 18 682 (78.6) | 9880 (81.2) | 8802 (75.8) |

Continued

**Table 2** Continued

| Characteristics | Total | Boys | Girls |
|---|---|---|---|
| Sometimes | 3765 (16.7) | 1653 (14.3) | 2112 (19.3) |
| Often/always | 1096 (4.7) | 535 (4.5) | 561 (4.9) |
| Suicide attempt | | | |
| Yes | 854 (3.6) | 337 (2.9) | 517 (4.4) |
| No | 22 689 (96.4) | 11 731 (97.1) | 10 958 (95.6) |
| Current smoking | | | |
| Yes | 1247 (5.5) | 968 (8.2) | 279 (2.5) |
| No | 22 296 (94.5) | 11 100 (91.8) | 11 196 (97.5) |
| Fighting | | | |
| Yes | 3707 (15.6) | 2966 (24.1) | 741 (6.4) |
| No | 19 836 (84.4) | 9102 (75.9) | 10 734 (93.6) |
| Being bullied | | | |
| Yes | 3041 (13.2) | 1872 (15.7) | 1169 (10.5) |
| No | 20 502 (86.8) | 10 196 (84.3) | 10 306 (89.5) |
| Sexual experience | | | |
| Yes | 861 (3.9) | 591 (5.1) | 270 (2.5) |
| No | 22 682 (96.1) | 11 477 (94.9) | 11 205 (97.5) |

Number in brackets were weighted proportion.

(27.0% vs 18.2%), among students who ever attempted suicide (48.4% vs 21.8% in those without suicide attempt), among current smokers (73.3% vs 19.8% in non-smokers), and among students who reported ever engaging in a physical fight (40.3% vs 19.5% in those without physical fight), ever being bullied (31.1% vs 21.5% in those without being bullied) or having sexual experience (59.1% vs 21.3% in those without sexual experience). There was no statistically significant difference between urban and rural areas (24.4% vs 22.0%). In addition, the prevalence of current drinking increased with poorer academic performance, longer duration of screen-time and higher severity of loneliness or insomnia (all P values for trend <0.0001).

The prevalence of binge drinking was 9.2% (95% CI 8.5 to 10.0) overall, and was 6.3%, 7.7% and 17.9%, respectively in middle school, academic high school and vocational high school. The associations of binge drinking with sociodemographic and behavioural factors were similar to those of current drinking. The prevalence of binge drinking was higher among boys than among girls (11.8% vs 6.5%), among students with previous suicide attempts (31.5% vs 8.4%), among students who reported smoking cigarettes (51.2% vs 6.8%), involvement in a physical fight (21.5% vs 7.0%), being bullied (14.1% vs 8.5%), previous sexual experience (39.2% vs 8.1%), poor academic performance, longer duration of screen-time and higher severity of loneliness or insomnia (P values for trend <0.0001 for the latter four parameters). There was no urban-rural difference in the prevalence of binge drinking (9.8% vs 9.0%).

### Logistic regression analysis

After adjusting for other variables included in the model, multivariable analysis showed that, compared with boys aged ≤13 years, older boys were more likely to binge drink (table 4). Compared with boys in middle school, boys attending academic high school and vocational high school had 1.5 (OR 1.48, 95% CI 1.15 to 1.90) and 2.1 (OR 2.09, 95% CI 1.59 to 2.74) times higher probability of binge drinking. Girls attending vocational high school were 2.2 times more likely to binge drink in comparison to girls attending middle school (OR 2.18, 95% CI 1.49 to 3.19). Boys with poor academic performance were 1.3 times more likely to binge drink in comparison to boys with excellent academic performance (OR 1.27, 95% CI 1.03 to 1.56). Compared with boys who were not physically active within the past 7 days, boys who were physically active on 6–7 days had a higher risk of binge drinking (OR 1.33, 95% CI 1.07 to 1.66). Compared with girls with screen-time of 0, girls with screen-time of 1–4 hours and >4 hours/day were 1.6 times (OR 1.61, 95% CI 1.20 to 2.17) and 2.6 (OR 2.59, 95% CI 1.93 to 3.48) times, respectively, more likely to binge drink. Boys with screen-time >4 hours/day had a higher risk of binge drinking in comparison to boys with screen-time of 0 (OR 1.44, 95% CI 1.16 to 1.79). Girls who often or always felt lonely were 1.4 times more likely to binge drink than girls who never or occasionally felt lonely (OR 1.43, 95% CI 1.05 to 1.95). Boys who ever attempted suicide had 2.4 times higher odds of binge drinking than boys without a history of attempted suicide (OR 2.37, 95% CI 1.73 to 3.25). The corresponding OR for girls was 3.4 (OR 3.36, 95% CI 2.48 to 4.56). Both boys and girls who reported smoking cigarettes within the past 30 days were 5.2 times (OR 5.21, 95% CI 4.13 to 6.58) and 6.7 (OR 6.68, 95% CI 4.64 to 9.60) times, respectively, more likely to binge drink than their counterparts who did not smoke cigarettes. Boys who reported being involved in physical fight in previous 12 months had a 2.2 times higher risk of binge drinking than those who reported not being involved in a physical fight (OR 2.18, 95% CI 1.81 to 2.62). The corresponding figure for girls was 2.5 (OR 2.48, 95% CI 1.87 to 3.29). Boys who had been bullied within the previous 12 months had a 1.2 times higher risk of binge drinking than boys who had not been bullied (OR 1.23, 95% CI 1.03 to 1.46). Both boys and girls who had sexual experience were 2.3 times (OR 2.33, 95% CI 1.69 to 3.20) and 1.8 (OR 1.82, 95% CI 1.22 to 2.72) times, respectively, more likely to binge drink than their counterparts without sexual experience.

### DISCUSSION

In this study of middle and high school students in Zhejiang, China, we examined the prevalence of current drinking and binge drinking, and identified and quantified the associations of sociodemographic and behavioural correlates of binge drinking, providing information to enable development of interventions to prevent binge drinking among this population group.

### Prevalence of binge drinking

Due to different definitions of binge drinking, direct comparisons between studies are difficult. Since using a

**Table 3** Weighted prevalence of current drinking and binge drinking by different subgroups

| Characteristics | Current drinking | | | Binge drinking | | |
|---|---|---|---|---|---|---|
| | Prevalence (%)* | $\chi^2$ | P value | Prevalence (%)* | $\chi^2$ | P value |
| Sex | | 93.65† | <0.0001 | | 80.46† | <0.0001 |
| Boys | 27.0 (25.6–28.5) | | | 11.8 (10.8–12.9) | | |
| Girls | 18.2 (16.8–19.6) | | | 6.5 (5.6–7.3) | | |
| Area | | 2.38† | 0.12 | | 0.62† | 0.4314 |
| Urban | 24.4 (21.9–26.9) | | | 9.8 (8.2–11.4) | | |
| Rural | 22.0 (20.5–23.5) | | | 9.0 (8.0–10.0) | | |
| Types of school | | 319.93† | <0.0001 | | 278.29† | <0.0001 |
| Middle school | 17.5 (16.2–18.8) | | | 6.3 (5.6–7.0) | | |
| Academic high school | 22.1 (20.4–23.8) | | | 7.7 (6.8–8.7) | | |
| Vocational high school | 35.9 (33.7–38.0) | | | 17.9 (15.9–20.0) | | |
| Academic performance | | 68.9‡ | <0.0001 | | 51.7‡ | <0.0001 |
| Excellent | 18.2 (16.4–20.0) | | | 6.9 (5.7–8.1) | | |
| Middle | 21.6 (20.2–22.9) | | | 8.5 (7.6–9.3) | | |
| Poor | 28.8 (26.9–30.6) | | | 12.7 (11.5–13.8) | | |
| Physical activity (days/week) | | 1.10‡ | 0.2977 | | 2.0‡ | 0.1555 |
| 0 | 23.2 (21.6–24.8) | | | 9.7 (8.7–10.8) | | |
| 1–2 | 22.0 (20.5–23.5) | | | 8.5 (7.4–9.5) | | |
| 3–5 | 21.7 (20.1–23.2) | | | 8.3 (7.3–9.3) | | |
| 6–7 | 25.0 (23.0–26.9) | | | 11.2 (9.9–12.5) | | |
| Screen-time (hour(s)/day) | | 330.9‡ | <0.0001 | | 259.6‡ | <0.0001 |
| 0 | 16.1 (14.8–17.3) | | | 5.5 (4.8–6.3) | | |
| <1 | 16.8 (15.3–18.3) | | | 5.5 (4.7–6.4) | | |
| 1–4 | 23.3 (21.8–24.7) | | | 8.9 (7.8–9.9) | | |
| ≥4 | 36.1 (34.0–38.1) | | | 17.9 (16.3–19.6) | | |
| Loneliness | | 149.6‡ | <0.0001 | | 113.9‡ | <0.0001 |
| Never/occasional | 19.6 (18.4–20.7) | | | 7.5 (6.8–8.2) | | |
| Sometimes | 26.2 (24.4–27.9) | | | 10.4 (9.4–11.5) | | |
| Often/always | 33.5 (31.4–35.6) | | | 16.5 (14.7–18.3) | | |
| Insomnia | | 177.2‡ | <0.0001 | | 205.3‡ | <0.0001 |
| Never/occasional | 20.4 (19.3–21.5) | | | 7.7 (7.0–8.3) | | |
| Sometimes | 29.3 (27.3–31.2) | | | 13.0 (11.6–14.5) | | |
| Often/always | 39.2 (35.8–42.5) | | | 22.7 (19.9–25.5) | | |
| Suicide attempt | | 204.61† | <0.0001 | | 288.99† | <0.0001 |
| Yes | 48.4 (44.2–52.6) | | | 31.5 (27.4–35.6) | | |
| No | 21.8 (20.6–23.0) | | | 8.4 (7.7–9.2) | | |
| Current smoking | | 7335.31† | <0.0001 | | 3720.33† | <0.0001 |
| Yes | 73.3 (70.5–76.0) | | | 51.2 (47.2–55.2) | | |
| No | 19.8 (18.9–20.8) | | | 6.8 (6.3–7.4) | | |
| Fighting | | 257.16† | <0.0001 | | 307.89† | <0.0001 |
| Yes | 40.3 (37.2–43.4) | | | 21.5 (19.1–23.9) | | |
| No | 19.5 (18.5–20.6) | | | 7.0 (6.3–7.6) | | |
| Being bullied | | 49.13† | <0.0001 | | 38.81† | <0.0001 |
| Yes | 31.1 (28.1–34.1) | | | 14.1 (12.0–16.2) | | |
| No | 21.5 (20.4–22.6) | | | 8.5 (7.8–9.2) | | |
| Sexual experience | | 526.80† | <0.0001 | | 500.40† | <0.0001 |
| Yes | 59.1 (54.5–63.6) | | | 39.2 (33.8–44.5) | | |
| No | 21.3 (20.2–22.4) | | | 8.1 (7.4–8.7) | | |

*Based on the weighted data.
†Rao-Scott $\chi^2$.
‡Trend for $\chi^2$.

**Table 4** Crude and adjusted OR of factors associated with binge drinking among adolescents in China

| Variable | Boys (n=12 068) | | Girls (n=11 475) | |
|---|---|---|---|---|
| | COR (95% CI) | AOR (95% CI) | COR (95% CI) | AOR (95% CI) |
| **Age groups (ref: ≤13 years)** | | | | |
| 14 | **1.95 (1.52 to 2.50)†** | **1.95 (1.51 to 2.53)†** | 1.12 (0.77 to 1.65) | 1.16 (0.76 to 1.77) |
| 15 | **2.70 (2.06 to 3.53)†** | **2.21 (1.67 to 2.90)†** | **1.46 (1.01 to 2.12)\*** | 1.12 (0.74 to 1.69) |
| ≥16 | **3.89 (3.07 to 4.92)†** | **2.10 (1.55 to 2.84)†** | **1.68 (1.13 to 2.50)\*\*** | 1.03 (0.60 to 1.77) |
| **Rural (ref: urban)** | 1.00 (0.77 to 1.29) | | 0.77 (0.57 to 1.04) | |
| **Types of school (ref: middle school)** | | | | |
| Academic high school | **1.62 (1.33 to 1.96)†** | **1.48 (1.15 to 1.90)\*\*** | 0.78 (0.60 to 1.00) | 0.99 (0.64 to 1.55) |
| Vocational high school | **3.65 (2.91 to 4.58)†** | **2.09 (1.59 to 2.74)†** | **2.71 (2.07 to 3.55)†** | **2.18 (1.49 to 3.19)\*\*\*** |
| **Parental marital status (ref: married)** | | | | |
| Others | **1.32 (1.08 to 1.61)\*\*** | 0.91 (0.73 to 1.13) | **2.07 (1.67 to 2.57)†** | 1.25 (0.95 to 1.65) |
| **Paternal education level** | | | | |
| **(ref: middle or below)** | | | | |
| High school | 1.00 (0.84 to 1.20) | 1.13 (0.93 to 1.38) | 0.84 (0.68 to 1.05) | 1.01 (0.80 to 1.27) |
| College or above | **0.76 (0.60 to 0.95)\*** | 0.94 (0.75 to 1.18) | **0.66 (0.49 to 0.90)\*\*** | 0.86 (0.62 to 1.19) |
| Unknown | 0.91 (0.72 to 1.16) | 0.99 (0.78 to 1.26) | 1.23 (0.89 to 1.71) | 1.25 (0.88 to 1.79) |
| **Maternal education level** | | | | |
| **(ref: middle or below)** | | | | |
| High school | 0.99 (0.84 to 1.17) | | 0.99 (0.77 to 1.27) | |
| College or above | 0.78 (0.61 to 1.00) | | 0.77 (0.56 to 1.05) | |
| Unknown | 0.82 (0.64 to 1.06) | | 1.11 (0.77 to 1.61) | |
| **Siblings (ref: no)** | 0.92 (0.80 to 1.06) | | 0.99 (0.80 to 1.22) | |
| **Academic performance (ref: excellent)** | | | | |
| Middle | **1.23 (1.01 to 1.50)\*** | 1.13 (0.93 to 1.36) | 1.28 (0.95 to 1.73) | 1.17 (0.87 to 1.58) |
| Bad | **1.82 (1.49 to 2.22)†** | **1.27 (1.03 to 1.56)\*** | **1.97 (1.43 to 2.70)†** | 1.30 (0.94 to 1.79) |
| **Physical activity (ref: 0 day/week)** | | | | |
| 1–2 days/week | 0.82 (0.66 to 1.02) | 0.90 (0.70 to 1.16) | 0.86 (0.70 to 1.06) | |
| 3–5 days/week | **0.79 (0.65 to 0.96)\*** | 0.93 (0.75 to 1.14) | 0.78 (0.60 to 1.02) | |
| 6–7 days/week | 1.09 (0.90 to 1.33) | **1.33 (1.07 to 1.66)\*** | 0.95 (0.74 to 1.21) | |
| **Screen-time (ref: 0 hour/day)** | | | | |
| <1 hour/day | 0.90 (0.72 to 1.12) | 0.94 (0.75 to 1.17) | 1.27 (0.88 to 1.84) | 1.18 (0.81 to 1.73) |
| 1–4 hours/day | **1.47 (1.19 to 1.82)\*\*\*** | 1.14 (0.91 to 1.43) | **2.13 (1.62 to 2.80)†** | **1.61 (1.20 to 2.17)\*\*** |
| ≥4 hours/day | **2.79 (2.28 to 3.41)†** | **1.44 (1.16 to 1.79)\*\*\*** | **5.91 (4.53 to 7.71)†** | **2.59 (1.93 to 3.48)†** |
| **Loneliness (ref: never/occasionally)** | | | | |
| Sometimes | **1.42 (1.23 to 1.63)†** | 1.08 (0.91 to 1.28) | **1.69 (1.37 to 2.08)†** | 1.23 (0.96 to 1.57) |
| Often/always | **2.25 (1.87 to 2.72)†** | 1.18 (0.90 to 1.56) | **3.14 (2.51 to 3.93)†** | **1.43 (1.05 to 1.95)\*** |
| **Insomnia (ref: never/occasionally)** | | | | |
| Sometimes | **1.97 (1.68 to 2.32)†** | **1.41 (1.13 to 1.77)\*\*** | **1.92 (1.59 to 2.31)†** | **1.33 (1.06 to 1.65)\*\*\*** |
| Often/always | **2.89 (2.29 to 3.64)†** | **1.69 (1.23 to 2.31)\*\*** | **5.18 (4.05 to 6.62)†** | **2.23 (1.63 to 3.06)†** |
| **Suicide attempt (ref: no)** | **4.14 (3.24 to 5.30)†** | **2.37 (1.73 to 3.25)†** | **7.35 (5.51 to 9.81)†** | **3.36 (2.48 to 4.56)†** |
| **Current smoking (ref: no)** | **11.24 (9.13 to 13.84)†** | **5.21 (4.13 to 6.58)†** | **20.27 (15.32 to 26.83)†** | **6.68 (4.64 to 9.60)†** |
| **Fighting (ref: no)** | **2.78 (2.37 to 3.27)†** | **2.18 (1.81 to 2.62)†** | **5.07 (4.04 to 6.37)†** | **2.48 (1.87 to 3.29)†** |
| **Being bullied (ref: no)** | **1.59 (1.34 to 1.89)†** | **1.23 (1.03 to 1.46)\*** | **1.81 (1.27 to 2.59)\*\*** | 1.04 (0.69 to 1.56) |
| **Sexual experience (ref: no)** | **6.92 (5.33 to 8.98)†** | **2.33 (1.69 to 3.20)†** | **6.60 (4.94 to 8.83)†** | **1.82 (1.22 to 2.72)\*\*** |

Bold numbers represent significant results.
AOR is adjusted for all other covariates in the model.
*P<0.05, **P<0.01, ***P<0.001, †P<0.0001.
COR, crude OR; AOR, adjusted OR.

5-drink measure for high school students in 1975, most national surveys have defined binge drinking as 5 or more drinks among both women and men.[12 17] A new gender-specific measure of ≥4/5 drinks for women/men has been used by Harvard School to reflect gender differences in the risk of alcohol-related harms.[30–32] The use of these gender-specific thresholds is also justified by women's generally smaller stature, and physiological differences between men and women affecting the absorption and distribution of alcohol.[33] Other studies have also used a 6-drink measure for both women and men.[18] In the present study, we adopted the definition of ≥4/5 drinks for women/men and found an overall prevalence of binge drink in Zhejiang of 9.2%, which was higher than previously reported in Hong Kong (7.1%).[15] The higher prevalence among boys than girls was consistent with results from other studies,[15 18 34] but differs from a Korean study, in which no sex difference was observed.[14] The highest prevalence of binge drinking in the current study was among students attending vocational high schools. A possible explanation for this was that, compared with middle school and academic high school students, vocational high school students are likely to enter employment immediately following graduation, for which social communication and interaction, possibly associated with alcohol consumption, may be considered important.[17] In subgroup analyses, the highest prevalence of binge drinking was among current drinkers (51.2%), students with sexual experience (39.2%) and students who had a history of a previous suicide attempt (31.5%). This suggests that such students should be identified as target populations for interventions to prevent and address binge drinking.

### Association of demographic factors of binge drinking

No association was found between parental educational levels and binge drinking in our study. In a systematic review, including 20 studies from 10 countries or areas, parental socioeconomic status, defined as the educational level, income or occupation, was weakly positively associated with binge drinking in low-income and middle-income countries. However, no such association was not found in high-income countries.[35] A previous study reported that non-intact family structure was associated with alcohol drinking among adolescents due to low family attachment or insufficient parent-child communication.[24 25] However, no such association was found in our study, which may reflect somewhat cultural differences in parental attitudes to alcohol consumption. van den Eijnden et al[36] found that adolescents were less likely to drink under strict alcohol-specific rules at homes despite non-intact families.

### Association of behavioural factors of binge drinking

A cohort study, including 89 university students who were followed up for 2 years, found persistent binge drinking was associated with verbal memory and monitoring difficulties.[37] This might be a possible reason for the positive association we observed between poor academic performance and binge drinking (ie, binge drinking may cause poor academic performance). Several earlier studies have also documented poorer performance among binge drinking students on neuropsychological tasks assessing inhibitory control, cognitive interference, sustained attention, verbal working memory and episodic declarative memory,[38–41] functions which are known to be supported by prefrontal and/or hippocampal regions, and these may also explain associations between poor academic performance and binge drinking.

Buscemi et al[42] found a positive relationship between moderate physical activity and alcohol use among males, but not females. Another study documented a positive relationship between vigorous physical activity and alcohol use, which was stronger at younger ages.[43] Our study showed that boys who were physically active were 1.3 times more likely to binge drink than boys who were not physically active. It is unclear whether physical activity leads to increased odds of binge drink or whether the converse is true (ie, binge drinking causes increased physical activity). One possible hypothesis might be that more physically active boys may binge drink as a means of relaxation. Further prospective studies are warranted to ascertain the likely direction of association between physical activity and binge drinking, and underlying mechanisms.

In our study, nearly one in five students reported >4 hours screen-time per day. Previous studies have mainly focused on television watching as a potential risk factor for various diseases.[44–47] However, the exponential growth of electronic screen products suggests that focusing only on television viewing might underestimate screen-time.[48] To the best of our knowledge, this is the first study to examine the association between screen-time and adolescent binge drinking, finding a positive association between duration of exposure to electronic screen products and odds of binge drinking.

Alcohol drinking can damage neurons, decrease neurogenesis, and cause cognitive and affective dysfunction, especially among adolescents.[49] Laboratory evidence has shown that decreased neurogenesis results in depression-like behaviours in rats.[50 51] The relationship of alcohol use with mental health has been suggested to be bidirectional. Rohde et al[52] reported that the onset of psychiatric disorder preceded the onset of alcoholism, while Berglund and Ojehagen[53] reported that depression followed alcoholism. However, although a previous study has indicated that substance use might be a means of self-medication and of alleviating the negative feelings that emanate from being lonely,[54] binge drinking has been shown to be ineffective in improving mental health.[55] In our study, loneliness was positively associated with adolescent binge drinking, but this association was found only among girls, consistent with findings among Arkhangelsk adolescents.[56] Furthermore, Huang et al[57] reported that alcohol drinking was significantly associated with emotional symptoms among girls only. Our findings suggest that preventive strategies against binge

drinking for female adolescents should include mental health consultation.

As demonstrated in our study, with higher levels of insomnia, the odds of adolescent binge drinking was higher. A previous study by Popovici and French,[58] including 14 089 participants, found that binge drinking was positively associated with sleep problems (having trouble falling asleep or staying asleep), independent of psychiatric conditions. There was a dose-response relationship between sleep problems and frequency of binge drinking, consistent with our study. In addition, Sharma et al[59] found that binge drinking could reverse sleep-wake cycle in rats and produce symptoms of insomnia, which could provide some explanation for the association between insomnia and binge drinking among students in the current study.

Alcohol drinking is a well-established risk factor for suicide attempts,[60 61] and acute intoxication may be a stronger risk factor than chronic alcohol use,[60] suggesting that binge drinking, which may produce rapid intoxication, may elevate the risk of suicide. In our study, prior suicide attempt was positively associated with binge drinking. Consistent with previous studies, binge drinking was related to the use of cigarettes.[62 63] In our study, current smoking had the strongest association with the odds of binge drinking, with over 5 and 6 times higher odds of binge drinking among current smoking boys and girls, respectively. Adolescents who drink alcohol are more likely to be involved in interpersonal conflicts and violence.[23] Our findings that both fighting and being bullied were positively related to adolescent binge drinking highlights the need for adolescent violence prevention programmes focusing on the reduction of alcohol abuse. We found a positive association between binge drinking and sexual experience, which was consistent with an earlier study.[23] Lewis et al[64] reported that alcohol may facilitate formation of intimate relationship. In addition, a previous study reported that binge drinking among adolescents was associated with higher rates of unwanted pregnancy, sexually transmitted infections and infertility.[8]

## Limitations

Our study had several limitations. First, the a cross-sectional study design prevents establishment of the causal relationships between sociodemographic and behavioural factors and binge drinking. Second, all data were self-reported by students, and self-reported alcohol consumption may be susceptible to recall and social desirability biases. Third, only students attending schools participated in the survey. Students who had been expelled or suspended from school, or who stopped attending, may be more likely to binge drink, and the overall prevalence of binge drinking in our current study might therefore represent an underestimate of the true prevalence.

## CONCLUSIONS

Despite these limitations, our study identified correlates of binge drinking among middle and high school students in Zhejiang and quantified the strength of

these associations, providing insight to inform binge drinking prevention strategies. Efforts to prevent binge drinking may need to address a cluster of correlating factors, including cigarette smoking, excessive screen-time, suicide attempt, fighting, being bullied, loneliness, insomnia and sexual behaviour. The presented findings provide evidence to assist healthcare providers in identifying students at high risk of binge drinking, which will aid in the planning of prevention and intervention measures for at-risk students.

**Acknowledgements** The authors would like to thank all the students, parents, teachers and local officials for their participation, assistance and cooperation.

**Contributors** HW designed the study, and collected and analysed the data with MY. JZ and RH were involved in data interpretation. HD and BF took part in data analysis and revised the manuscript. MW was involved in data collection. All the authors have read and approved the final submitted version.

**Funding** The work was supported by grant (2016YFC0900502) from National Key Research and Development Program of China.

**Competing interests** None declared.

**Patient consent** Detail has been removed from this case description/these case descriptions to ensure anonymity. The editors and reviewers have seen the detailed information available and are satisfied that the information backs up the case the authors are making.

**Ethics approval** Ethics Committee of Zhejiang Provincial Centre for Disease Control and Prevention.

**Provenance and peer review** Not commissioned; externally peer reviewed.

**Data sharing statement** No additional data are available.

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
