## [Reviewer comments · BMJ Open]

ARTICLE DETAILS

TITLE (PROVISIONAL)	Binge Drinking and Associated Factors among School Students: A Cross-sectional Study in Zhejiang Province, China
AUTHORS	Wang, Hao; Hu, Ru-Ying; Zhong, Jie-Ming; Du, Huaidong; Bragg, Fiona; Wang, Meng; Yu, Min

VERSION 1 – REVIEW

REVIEWER	Guoqing Hu Central South University
REVIEW RETURNED	19-Dec-2017

GENERAL COMMENTS	This paper reports valuable information on binge drinking among Chinese adolescents based on a large sample survey. But this paper could be improved in the following issues. 1. It is good to replace adolescent with school students and specify the place of study "Zhejiang Province" in the title.2. Rows 5-6 on page 3: This sentence is not readable for "and correlates".3. Rows 9-12 on page 3: Please specify the category of schools. Key outcome measures need to be defined and mentioned briefly.4. Rows 14-22 on page 3: Please describe the sample characteristics briefly and include 95% CI of key measures. "screen time" is hard to understand. Please improve the wording for it.5. Rows 24-28 on page 3: Please limit the conclusions to school students since the adolescents out of schools were excluded from this study. The implication of analysis for influencing factors should be mentioned. Row 12 on page 4: "temporal" may not be a proper word. Probably, use "causal" to replace it. Rows 14-17 on page 4: "school-going" is hard to read. The last sentence should be improved since there is some evidence to support it. Rows 14-32 on page 5: The generation of research question of this study is weak and should be improved to emphasize that what knowledge gaps that this study would fill up. Rows 37-54 on page 5: Please clearly define study subjects of this study. Especially, please define "middle school", "academic high
--

	schools". It is very hard for me to understand their differences. Do the authors mean "junior high school and senior high schools"? Also, please classify the determination of sample size. Rows 1-18 on page 6: Please include the selection of covariates and their operational definitions. Rows 36-52 on page 6: Statistical analysis are generally good. But the school clustering probably exists for the same schools, it may be better to use multi-level statistical models for data analysis. In addition, please briefly mention why they conducted analyses by sex separately. Rows 12-18 on page 7: I am not sure why the authors conducted sample characteristics by sex. Rows 22-30 on page 7: Please report proportions of drinking and binge drinking by key covariates here. Rows 35-55 on page 7: results of univariate analysis could be omitted in main text. It looks that the presentation of logistic regression is incomplete; some significant factors were not mentioned in text. I want to know why. Table 1: Key indicators should be included in the main text. Table 2: Please correct small errors in Table 2, e.g. missing a space between 6.5 and 95% CI. Table 3: Chi-square tests seem to make little sense to the results. Pages 8-9: Generally, the discussion is hard to read. If possible, please first summarize key findings of this study and then discuss them with the published literature point by point. It would be great if the authors add subtitles within the discussion and emphasize the new contribution of this study by comparing their results with the known knowledge.
--	---

REVIEWER	Marina Bosque-Prous Agència de Salut Pública de Barcelona, Spain
REVIEW RETURNED	02-Jan-2018

GENERAL COMMENTS	The article entitled "Binge drinking and associated factors among adolescent students in China" aims at estimating the prevalence of binge drinking in Chinese adolescents and the individual factors associated with it. It is a potentially valuable piece of work. However, the following issues need to be addressed: Abstract: - The authors should add that these study was a cross-sectional study using data from a school-based survey. Introduction: - The authors should include some information about other studies conducted in China or other countries in Asia, as probably their drinking culture is more similar between them than the ones in Europe. Some alcohol use prevalence should be added to help contextualize. - It should help the reader if the authors include some information about the Zhejiang Province. Does this region represent the average
--

region in China according to socioeconomic and demographic characteristics? Besides, in the discussion section the authors should justify and provide some data about the representativeness of this sample from the adolescent population of China.

Methods:

- The authors explain how the sampling design was made, but it is not clear whether the sampling of the 30 counties was random or a convenience sampling. Please clarify.
- Given the high prevalence of current drinking and binge drinking, it is advisable to use prevalence ratio (PR) instead of odds ratio (OR). In these cases, OR tends to overestimate the associations and PR are easily interpreted. Nowadays, several statistical software can estimate PR. I encourage the authors to reanalyze the data to estimate the associations of the different variables with binge drinking using PR. The following articles provide further information on this suggestion:

Barros AJD, Hiraakata VN. Alternatives for Logistic Regression in Cross-Sectional Studies: An Empirical Comparison of Models That Directly Estimate the Prevalence Ratio. *BMC Medical Research Methodology*. 2003;3: 21.

Espelt A, Mari-Dell'Olmo M, Penelo E, Bosque-Prous M. Applied Prevalence Ratio Estimation with Different Regression Models: An Example from a Cross-National Study on Substance Use Research. *Adicciones*. 2017;2:29.

Results:

- Table 2 should be the characteristics of the sample instead of the prevalence of alcohol use. I suggest changing the order of the table 2 and 3.

Discussion:

- The authors compare the findings mainly with European studies. There are some studies in China and other Asiatic countries which estimate alcohol use prevalence and some associated factors that should be of interest for discussing the results of the study, such as your references 18, 19, 20. But there are also other articles that may be useful, for example:

Huang R, Ho SY, Wang MP, Lo WS, Lam TH. Reported alcohol drinking and mental health problems in Hong Kong Chinese adolescents. *Drug Alcohol Depend*. 2016 Jul 1;164:47-54.

Kim JH, Chan KW, Chow JK, Fung KP, Fong BY, Cheuk KK, Griffiths SM. University binge drinking patterns and changes in patterns of alcohol consumption among Chinese undergraduates in a Hong Kong university. *J Am Coll Health*. 2009 Nov-Dec;58(3):255-65.

Kim JH, Lee S, Chow J, Lau J, Tsang A, Choi J, Griffiths SM. Prevalence and the factors associated with binge drinking, alcohol abuse, and alcohol dependence: a population-based study of Chinese adults in Hong Kong. *Alcohol Alcohol*. 2008 May-Jun;43(3):360-70.

- There are other recent articles that estimate prevalence of binge drinking in China and analyze possible related factors. It is important that the authors state more clearly what this study adds to the current literature.

- In the limitations section, the authors should add the social desirability bias, associated with self-reported data. Some people, especially the ones drinking more, tend to underreport their drinking.

- Some conclusions of the study should be added in the last paragraph of the discussion section.

	Tables:  - In table 3, the results presented should also be weighted as in table 2. Besides, the proportion in some variables is presented using one decimal and in other using two. - In table 4, the adjusted model should only include the variables that seemed to be associated with binge drinking in the crude model and whose relationship continued statistically significant after adjusting for the other variables. Unless these variables are confounders on the model
--	--

VERSION 1 – AUTHOR RESPONSE

Responses to the Journal requirements

1. Please include the study design in the title.

Response: Done. The title has been revised as follows “Binge Drinking and Associated Factors among School Students: A Cross-sectional Study in Zhejiang Province, China”.

2. Please include the response rate in the abstract.

Response: Done. Response rate was supplemented in the abstract section.

Responses to Reviewer #1

1. It is good to replace adolescent with school students and specify the place of study "Zhejiang Province" in the title.

Response: Done. Adolescent students were replaced by school students, and the place of study "Zhejiang Province" was added in the title.

2. Rows 5-6 on page 3: This sentence is not readable for "and correlates".

Response: Done. This sentence was revised as follows “To investigate the prevalence and correlating factors of binge drinking among middle and high school students in Zhejiang, China”.

3. Rows 9-12 on page 3: Please specify the category of schools. Key outcome measures need to be defined and mentioned briefly.

Response: Thanks! The category of schools was included in abstract section. The definition of binge drink was added briefly as follows“(i.e., drinking four or more alcoholic drinks for girl in a row of 1 to 2 hours and five or more for boy)”.

4. Rows 14-22 on page 3: Please describe the sample characteristics briefly and include 95% CI of key measures. "screen time" is hard to understand. Please improve the wording for it.

Response: Good suggestions! The sample characteristics were described briefly. 95% CI of key measures, including current drink and binge drink, were provided in abstract. "Screen time" was replaced by "excessive exposure to electronic screen products".

5. Rows 24-28 on page 3: Please limit the conclusions to school students since the adolescents out of schools were excluded from this study. The implication of analysis for influencing factors should be mentioned.

Response: Thanks! “Adolescents” was replaced by “middle and high school students”. The implication was added in the conclusion.

6. Row 12 on page 4: "temporal" may not be a proper word. Probably, use "causal" to replace it.

Response: Thanks! Word "causal" was replaced by word "temporal".

Response: Done.

7. Rows 14-17 on page 4: "school-going" is hard to read. The last sentence should be improved since there is some evidence to support it.

Response: Thanks! "school-going adolescents" was replaced by "students attending schools". The last sentence was amended.

8. Rows 14-32 on page 5: The generation of research question of this study is weak and should be improved to emphasize that what knowledge gaps that this study would fill up.

Response: Thanks! The introduction section was rewritten.

9. Rows 37-54 on page 5: Please clearly define study subjects of this study. Especially, please define "middle school", "academic high schools". It is very hard for me to understand their differences. Do the authors mean "junior high school and senior high schools"? Also, please classify the determination of sample size.

Response: The content of subjects was added in "Survey design and participants" section. The sentence "In China, after 6-years education at primary schools, children usually attend middle schools (i.e., junior high school) for 3-years education (grades 7-9). After graduation from middle schools, they enter high schools (i.e., senior high schools, including academic high schools and vocational high schools) for another 3-years education (grades 10-12)." was supplemented to describe the types of school. Detailed information about sample size was provided in Method section as follows "The sample size was calculated by using the formula: $N = \frac{d^2 \times P \times (1-P)}{d^2}$. Means and 95% confidence interval (CI; 2-sided for $u=1.96$) were determined; the prevalence of binge drinking (10%) obtained in the China was used as a measure of probability (p)³⁴; the design effect (deff) value was set at 3; and the relative error was: $d = r \times 1\%$, $r = 15\%$. Based on these parameters, the sample size for each stratum was estimated to be 4610 subjects. Because there were 4 strata (Areas: urban and rural. Sex: boy and girl), and assuming a potential nonresponse rate of 20%, the final sample size was calculated as 23050".

10. Rows 1-18 on page 6: Please include the selection of covariates and their operational definitions.

Response: Thanks! Covariates and their operational definitions were added in main text as follows "Physical activity was assessed by the question: "During the past 7 days, on how many days were you physically active for a total of at least 60 minutes per day?". Answer options included: "None", "1 day", "2 days", "3 days", "4 days", "5 days", "6 days" and "7 days". Answers were further categorized into four groups: "None", "1-2 d/w", "3-5 d/w", and "6-7 d/w". Current smoking was assessed by the question: "During the past 30 days, on how many days did you smoke cigarettes?". Answer options included: "None", "1-2 days", "3-5 days", "6-9 days", "10-19 days", "20-29 days" and "all 30 days". Current smoking was defined as smoking cigarettes at least one day in the past 30 days. Screen time was collected through the question: "On an average school day, how many hours do you play video or computer games or use a computer for something that is not school work?". Answer options included: "None", "< 1 h/d", "1 h/d", "2 h/d", "3 h/d", "4 h/d", "≥ 5 h/d"). Answers were further categorized into four groups: "None", "<1 h/d", "1-4 h/d", "≥ 4 h/d". Suicidal attempt was assessed using the question: "During the past 12 months, how many times did you actually attempt suicide?". Response options included: "None", "1 time", "2-3 times", "4-5 times", "6 or more times". Suicide attempt was defined as attempting suicide at least one time in the past 12 months. Fighting was assessed by the question: "During the past 12 months, how many times were you in a physical fight?". Answer options included "None", "1 time", "2-3 times", "4-5 times", "6-7 times", "8-9 times", "10-11 times", "12 or more times". Fighting was defined physical fight at least one time in the past 12 months. Being bullied was assessed by the question: "During the past 12 months, how many times has someone threatened or injured you with a weapon such as a gun, knife, or club on school property?". Answer options included "None", "1 time", "2-3 times", "4-5 times", "6-7 times", "8-9 times", "10-11 times", "12 or more times". Being bullied was defined as being threatened or injured by someone at least one time in the past 12 months. Answer was dichotomized into "Yes" and "No". More detailed covariates information was provided in Table 1".

11. Rows 36-52 on page 6: Statistical analysis are generally good. But the school clustering probably exists for the same schools, it may be better to use multi-level statistical models for data analysis. In addition, please briefly mention why they conducted analyses by sex separately.

Response: Thanks for this suggestion. Statistical inferences in the present study were performed with design-based approach, which had already accounted for clustering (non-independence) of individuals from the perspective of complex sample design. Like multi-level modeling, the design-based approach is able to estimate correct sampling error in analysis of hierarchical data. In addition, design-based approach can attenuate sampling bias through post-stratification and non-response weighting." The following article provided further information on this suggestion:

S.G. Heeringa, B.T. West, P.A. Berglund, Applied Survey Data Analysis, CRC Press, Boca Raton, 2010.

12. Rows 12-18 on page 7: I am not sure why the authors conducted sample characteristics by sex. Response: Because some behaviors of boys differ from those of girls. For example, cigarettes smoking, alcohol drinking, fighting were more prevalent among boys than those among girls (even between adult Chinese men and women, smoking and drinking behaviors differ dramatically. For example Chen Z, Peto R, Zhou M, et al. Contrasting male and female trends in tobacco-attributed mortality in China: evidence from successive nationwide prospective cohort studies. *Lancet*, 2015, 386 (10002):1447-56.). Lacking of physical activity was more common among girls than that among boys.

13. Rows 22-30 on page 7: Please report proportions of drinking and binge drinking by key covariates here.

Response: Good suggestions! Prevalence and 95%CI of drinking and binge drinking by different key covariates, including physical activity, screen time, loneliness, insomnia, suicide attempt, current smoking, fighting, being bullied, sexual experience, was supplemented in Table 2.

14. Rows 35-55 on page 7: results of univariate analysis could be omitted in main text. It looks that the presentation of logistic regression is incomplete; some significant factors were not mentioned in text. I want to know why.

Response: Thanks! Results of univariate analysis were deleted from the main text. Key significant factors in multivariable logistic regression were added in main text as following: "After adjusted other variables in the model, multivariable analysis showed that compared to boys aged ≤ 13 years, boys with older age were more likely to binge drink. Compared to boys of middle school, boys of academic high school and vocational high school had 1.5 and 2.1 times more likely probability of binge drinking, respectively ((OR=1.48, 95%CI: 1.15-1.90) vs. OR=2.09, 95%CI: 1.59-2.74)). Girls of vocational high school were 2.2 times more likelihood of binge drinking in comparison to girls of middle school (OR=2.18, 95%CI: 1.49-3.19). Boys with poor academic performance were 1.3 times more likely to binge drink in comparison to boys with excellent academic performance (OR=1.27, 95%CI: 1.03-1.56). Compared to boys who were not physically active within the past 7 days, boys being physically active 6-7 days had a higher risk of binge drinking (OR=1.33, 95%CI: 1.07-1.66). Compared to girls who spent no time on playing electronic screen products, girls who spent 1-4 hours per day and more than 4 hours per day were 1.6 times and 2.6 times more likelihood of binge drinking, respectively. ((OR=1.61, 95%CI: 1.20-2.17) vs. (OR=2.59, 95%CI: 1.93-3.48)). Boys who spent more than 4 hours per day on electronic screen products had a higher risk of binge drinking in comparison to boys who spent no time on playing electronic screen products (OR=1.33, 95%CI: 1.07-1.66). Girls who often or always felt lonely were 1.4 times more likely to binge drink than girls who never or occasionally felt lonely (OR=1.43, 95%CI: 1.05-1.95). Boys ever attempted suicide had a 2.4 times increased odds of binge drinking than boys without committing suicide (OR=2.37, 95%CI: 1.73-3.25). The corresponding odds ratio of girls was 3.4. (OR=3.36, 95%CI: 2.48-4.56). Both boys and girls who smoked cigarettes within the past 30 days were 5.2 times and 6.7 times more likely to binge drink than their counterparts who did not smoke cigarettes, respectively ((OR=5.21, 95%CI: 4.13-6.58) vs. (OR=6.68, 95%CI: 4.64-9.60)). Boys ever involved in physical fight in past 12 months had a 2.2 times higher risk of binge drinking than those not in physical fights (OR=2.18, 95%CI: 1.73-3.25). The corresponding figure of girls was 2.5. (OR=2.48, 95%CI: 1.87-3.29). Boys ever being bullied within the past 12 months had a 1.2 times higher risk of binge drinking than boys not being bullied (OR=1.23, 95%CI: 1.03-1.46). Both boys and girls who had sexual experience were 2.3 times and 1.8 times more likely to binge drink than their counterparts without sexual experience, respectively ((OR=2.33, 95%CI: 1.69-3.20) vs. (OR=1.82, 95%CI: 1.22-2.72))."

15. Table 1: Key indicators should be included in the main text.

Response: Thanks for the reminding. Key indicators in table 1 were added in Method section in the main text.

16. Table 2: Please correct small errors in Table 2, e.g. missing a space between 6.5 and 95% CI.

Response: Thanks for the reminding. Error was corrected in Table 2.

17. Table 3: Chi-square tests seem to make little sense to the results.

Response: I agree with reviewer's opinion! Chi-square value and P value were deleted.

18. Pages 8-9: Generally, the discussion is hard to read. If possible, please first summarize key findings of this study and then discuss them with the published literature point by point. It would be great if the authors add subtitles within the discussion and emphasize the new contribution of this study by comparing their results with the known knowledge.

Response: Thanks! Discussion section has been amended according to reviewer's suggestions. Five subtitles were added in the Discussion section.

Responses to Reviewer #2

Abstract:

1. The authors should add that these study was a cross-sectional study using data from a school-based survey.

Response: Done.

Introduction:

2. The authors should include some information about other studies conducted in China or other countries in Asia, as probably their drinking culture is more similar between them than the ones in Europe. Some alcohol use prevalence should be added to help contextualize.

Response: Other studies, conducted in China and Korea, were added in introduction section.

3. It should help the reader if the authors include some information about the Zhejiang Province. Does this region represent the average region in China according to socioeconomic and demographic characteristics? Besides, in the discussion section the authors should justify and provide some data about the representativeness of this sample from the adolescent population of China.

Response: Information about the Zhejiang Province was added as following "Zhejiang province, standing in the east of China, has a population of 56 million. It has experienced rapid economic development in the past 30 years". Zhejiang is one of all of 30 provinces in China, and it cannot represent the average region in China. The result can only represent Zhejiang region. We revised the title as "Binge Drinking and Associated Factors among School Students: A Cross-sectional Study in Zhejiang Province, China".

Methods:

4. The authors explain how the sampling design was made, but it is not clear whether the sampling of the 30 counties was random or a convenience sampling. Please clarify.

Response: Thanks! 30 counties were sampled randomly from 90 counties. We revised the related description in the main text.

5. Given the high prevalence of current drinking and binge drinking, it is advisable to use prevalence ratio (PR) instead of odds ratio (OR). In these cases, OR tends to overestimate the associations and PR are easily interpreted. Nowadays, several statistical software can estimate PR. I encourage the authors to reanalyze the data to estimate the associations of the different variables with binge drinking using PR. The following articles provide further information on this suggestion:

Barros AJD, Hirakata VN. Alternatives for Logistic Regression in Cross-Sectional Studies: An Empirical Comparison of Models That Directly Estimate the Prevalence Ratio. *BMC Medical Research Methodology*. 2003;3: 21.

Espelt A, Mari-Dell'Olmo M, Penelo E, Bosque-Prous M. Applied Prevalence Ratio Estimation with Different Regression Models: An Example from a Cross-National Study on Substance Use Research. *Adicciones*. 2017;2:29.

Response: I agree with the reviewer that OR tends to overestimate the associations when the disease/condition under investigation is not rare (e.g. disease prevalence >30%). On the other hand, if it is a rare disease/condition, i.e. with prevalence lower than 10%, OR should be a valid approximate of PR. In our study, the prevalence of binge drinking is 9.2%, therefore should be acceptable to use OR in the place of PR in our study. In addition, statistical inferences in the present study were performed using design-based approach (PROC SURVEYLOGISTIC procedure in SAS) to take into account the complex survey sampling weights. It is hard to find similar approach to calculate PR.

Results:

6. Table 2 should be the characteristics of the sample instead of the prevalence of alcohol use. I suggest changing the order of the table 2 and 3.

Response: Thanks! It was done.

Discussion:

7. The authors compare the findings mainly with European studies. There are some studies in China and other Asiatic countries which estimate alcohol use prevalence and some associated factors that should be of interest for discussing the results of the study, such as your references 18, 19, 20. But there are also other articles that may be useful, for example:

Huang R, Ho SY, Wang MP, Lo WS, Lam TH. Reported alcohol drinking and mental health problems in Hong Kong Chinese adolescents. *Drug Alcohol Depend*. 2016 Jul 1;164:47-54.

Kim JH, Chan KW, Chow JK, Fung KP, Fong BY, Cheuk KK, Griffiths SM. University binge drinking patterns and changes in patterns of alcohol consumption among Chinese undergraduates in a Hong Kong university. *J Am Coll Health*. 2009 Nov-Dec;58(3):255-65.

Kim JH, Lee S, Chow J, Lau J, Tsang A, Choi J, Griffiths SM. Prevalence and the factors associated with binge drinking, alcohol abuse, and alcohol dependence: a population-based study of Chinese adults in Hong Kong. *Alcohol Alcohol*. 2008 May-Jun;43(3):360-70.

Response: Thanks! Discussion was divided into five sections and was rewritten. More references including three references you recommended and more detailed information were added in the discussion.

8. There are other recent articles that estimate prevalence of binge drinking in China and analyze possible related factors. It is important that the authors state more clearly what this study adds to the current literature.

Response: Thanks! Different definitions were added and introduced in discussion for better understanding. Due to different definition of binge drinking, it is difficult to compare different studies directly. We emphasize on comparing the prevalence of binge drinking with Hong Kong and Korea (the same definition). More subgroup analysis were added in the discussion, which are useful for the development of adolescent binge drinking prevention strategies targeting on vulnerable groups in Zhejiang.

9. In the limitations section, the authors should add the social desirability bias, associated with self-reported data. Some people, especially the ones drinking more, tend to underreport their drinking.

Response: THANKS! Social desirability bias was added in the limitations.

10. Some conclusions of the study should be added in the last paragraph of the discussion section.

Response: Good suggestions! Conclusions were added in the last paragraph of the discussion as follows: "Despite these limitations, our study identified the correlates of binge drinking among middle and high school students in Zhejiang and quantified their strength of associations, and provided information for preventive strategies against binge drinking. Efforts to prevent binge drinking may need to address a cluster of correlating factors, including cigarettes smoking, excessive exposure to electronic screen products, suicide attempt, fighting, being bullied, loneliness, insomnia, sexual behaviour, and so on. The findings would be instructive for healthcare providers to detect students with high-risk of binge drinking, which will aid in planning intervention measures for at-risk students."

Tables:

11. In table 3, the results presented should also be weighted as in table 2. Besides, the proportion in some variables is presented using one decimal and in other using two.

Response: Done. The results of basic characteristics were weighted. Variables were presented using one decimal.

12. In table 4, the adjusted model should only include the variables that seemed to be associated with binge drinking in the crude model and whose relationship continued statistically significant after adjusting for the other variables. Unless these variables are confounders on the model.

Response: The results of adjusted model were recalculated. We first determine which factors were associated with binge drinking in univariate analyses ($P < 0.05$). Variables significant in the univariate analyses were entered in a multivariate logistic regression model.

VERSION 2 – REVIEW

REVIEWER	Guoqing Hu Central South University, China
REVIEW RETURNED	31-Jan-2018

GENERAL COMMENTS	My comments are all addressed. Thanks. I strongly suggest the authors find a native English speaker to polish the writing of English language of this paper. Some expressions are hard to read although I could understand them.
---

REVIEWER	Marina Bosque-Prous Universitat Oberta de Catalunya
REVIEW RETURNED	16-Feb-2018

GENERAL COMMENTS	The manuscript has much improved. However, there is still some concern related to the analyses conducted. The authors stated that the prevalence of binge drinking was lower than 10% in their sample, but when looking at table 3, the prevalence of binge drinking is really high for some characteristics of the sample. For example, about 50% of current smokers also binge drank. Besides, in table 3, the authors estimated prevalences instead of odds, so it will be more coherent to calculate prevalence ratios to estimate the associations between variables. The authors should revise the manuscript to correct some language mistakes. For example, in the limitations section (line 34), they should change casual by causal.
--

VERSION 2 – AUTHOR RESPONSE

Responses to the Journal requirements

1. Please ensure to have the same CONTRIBUTORSHIP STATEMENT both in your main document and in Scholar One.

Response: Done.

2. We have implemented an additional requirement to all articles to include 'Patient and Public Involvement' statement within the main text of your main document. Please refer below for more information regarding this new instruction:

Authors must include a statement in the methods section of the manuscript under the sub-heading 'Patient and Public Involvement'.

This should provide a brief response to the following questions:

How was the development of the research question and outcome measures informed by patients' priorities, experience, and preferences?

How did you involve patients in the design of this study?

Were patients involved in the recruitment to and conduct of the study?

How will the results be disseminated to study participants?

For randomised controlled trials, was the burden of the intervention assessed by patients themselves?

Patient advisers should also be thanked in the contributorship statement/acknowledgements.

If patients and or public were not involved please state this.

Response: Done. "Patient and Public Involvement" was added in method section. Participants' parents were thanked in the Acknowledgements.

Responses to Reviewer #1

1. I strongly suggest the authors find a native English speaker to polish the writing of English language of this paper. Some expressions are hard to read although I could understand them.

Response: Done. The article was polished by a native English speaker named Bragg Fiona.

Responses to Reviewer #2

1. The manuscript has much improved. However, there is still some concern related to the analyses conducted. The authors stated that the prevalence of binge drinking was lower than 10% in their sample, but when looking at table 3, the prevalence of binge drinking is really high for some characteristics of the sample. For example, about 50% of current smokers also binge drank. Besides, in table 3, the authors estimated prevalences instead of odds, so it will be more coherent to calculate prevalence ratios to estimate the associations between variables.

Response: Thanks for this suggestion. In the present study, statistical inferences were performed using design-based approach (PROC SURVEYLOGISTIC procedure in SAS) to take into account the complex survey sampling weights. It is hard to find similar approach to calculate PR. The following article provided further information on this suggestion:

S.G. Heeringa, B.T. West, P.A. Berglund, Applied Survey Data Analysis, CRC Press, Boca Raton, 2010.

2. The authors should revise the manuscript to correct some language mistakes. For example, in the limitations section (line 34), they should change casual by causal.

Response: Done. The article was polished by a native English speaker named Bragg Fiona.